# Diagnostic testing preferences can help inform future public health response efforts: Global insights from an international survey

Leah Salzano[1,◐], Nithya Narayanan[1,◐], Emily R. Tobik[1,◐], Sumaira Akbarzada[1], Yanjun Wu[1], Sarah Megiel[2], Brittany Choate[1,3], Anne L. Wyllie[1]*

1 Department of Epidemiology of Microbial Diseases, Yale School of Public Health, New Haven, Connecticut, United States of America, 2 Department of Health Policy and Management, Yale School of Public Health, New Haven, Connecticut, United States of America, 3 SalivaDirect, Inc., New Haven, Connecticut, United States of America

◐ These authors contributed equally to this work.
* awyllie@gmail.com

**Data Availability Statement:** The data that support the findings of this study are available in the manuscript and in the Supporting Information.

## Abstract

Public perception regarding diagnostic sample types as well as personal experiences can influence willingness to test. As such, public preferences for specific sample type(s) should be used to inform diagnostic and surveillance testing programs to improve public health response efforts. To understand where preferences lie, we conducted an international survey regarding the sample types used for SARS-CoV-2 tests. A Qualtrics survey regarding SARS-CoV-2 testing preferences was distributed via social media and email. The survey collected preferences regarding sample methods and key demographic data. Python was used to analyze survey responses. From March 30th to June 15th, 2022, 2,094 responses were collected from 125 countries. Participants were 55% female and predominantly aged 25–34 years (27%). Education and employment were skewed: 51% had graduate degrees, 26% had bachelor's degrees, 27% were scientists/researchers, and 29% were healthcare workers. By rank sum analysis, the most preferred sample type globally was the oral swab, followed by saliva, with parents/guardians preferring saliva-based testing for children. Respondents indicated a higher degree of trust in PCR testing (84%) vs. rapid antigen testing (36%). Preferences for self- or healthcare worker-collected sampling varied across regions. This international survey identified a preference for oral swabs and saliva when testing for SARS-CoV-2. Notably, respondents indicated that if they could be assured that all sample types performed equally, then saliva was preferred. Overall, survey responses reflected the region-specific testing experiences during the COVID-19. Public preferences should be considered when designing future response efforts to increase utilization, with oral sample types (either swabs or saliva) providing a practical option for large-scale, accessible diagnostic testing.

**Funding:** The study was supported by SalivaDirect, Inc (ALW). The funders had no role in study design, data collection and analysis, decision to publish, or preparation of the manuscript. The study protocol was designed by the Yale researchers. The decision to publish was made by the Yale researchers; all authors agree with the decision to publish and with the results of the study.

**Competing interests:** I have read the journal's policy and the authors of this manuscript have the following competing interests: ALW has received consulting and/or advisory board fees from Pfizer, Merck, Diasorin, PPS Health, Co-Diagnostics, and Global Diagnostic Systems for work unrelated to this project, and is Principal Investigator on research grants from Pfizer, Merck, NIH RADx UP and SalivaDirect, Inc. to Yale University and from NIH RADx, Balvi.io and Shield T3 to SalivaDirect, Inc. All other co-authors declare no potential conflict of interest.

## Introduction

Strategies for diagnostic testing rapidly evolved during the coronavirus disease 2019 (COVID-19) pandemic in response to unprecedented worldwide demand. These additional tools are now available to help support seasonal respiratory virus epidemics or future outbreaks of emerging pathogens. Testing during outbreaks remains the most reliable tool for directing efforts and resources; it is the identification of infected individuals that can help to mitigate transmission [1]. The variety of test modalities available (including polymerase chain reaction [PCR], loop-mediated isothermal amplification [LAMP], antigen, and serology tests; **Table A in S1 Appendix**) can be used to support outbreak-response efforts in three key ways: 1). diagnostic testing for symptomatic individuals, 2). diagnostic testing for asymptomatic individuals, and 3). large-scale screening for ongoing disease surveillance.

At the start of the COVID-19 pandemic, testing defaulted to the use of nasopharyngeal or oropharyngeal swabs, which soon fell under supply chain constraints. Moreover, this placed a high demand on trained medical personnel, especially those who required full personal protective equipment (PPE) to collect these technical sample types. As the need for frequent, repeat testing was realized, simpler sample types became available, namely, nasal (anterior-nares or mid-turbinate) swabs and saliva.

Compared to swab-based approaches, various studies have shown that saliva sampling causes less discomfort for the patient, decreases the risk of viral transmission during testing, decreases the need for PPE, and reduces overall costs [2]. In contrast, nasopharyngeal and nasal swabs are recognized as invasive sample types, with discomfort from sample collection leading to testing aversion and decreased testing or screening rates, especially in situations requiring frequent or repeat testing. For example, when parents, their school-age children, and school staff were all offered COVID-19 tests through saliva sample collection and then asked whether they would have consented to testing if, instead, a nasal swab was offered, 36% reported they would have refused a nasal swab if saliva sampling was available as an option [2]. Another COVID-19 testing preference survey conducted in kindergarten to 12th grade (K-12) schools found that 51% of elementary students preferred saliva over a nasal swab because it was "easier and more fun" [3]. The opinions of school-age children and their parents highlight the impact of personal preferences for various sampling methods when making medical decisions. These findings offer valuable insights into the factors influencing people's willingness to undergo future diagnostic testing and can inform and shape future procedures and policies related to acceptable testing practices [2].

Despite the advancements in testing technologies for COVID-19, access and uptake of the different sample types has varied around the globe. Understanding the testing landscape within and between countries is essential for making fair comparisons about local infection rates and the "acceptability" of testing in different regions. Insight into public preferences surrounding diagnostic testing, screening and surveillance, such as those in the examples above, can also help inform test development and the design and implementation of testing programs that meet the needs of the target population. In addition, knowledge and opinions on testing options and sample types are essential for government and public health officials to best maximize their spending and impact [4]. Therefore, in the current study, we surveyed people's preferences from around the world regarding the clinical sample types used for the detection of severe acute respiratory syndrome coronavirus 2 (SARS-CoV-2, the virus that causes COVID-19) and the diagnosis of COVID-19. We aimed to better understand the public view regarding different sample types with the hope that these results could inform future surveillance planning and diagnostics development for not only COVID-19, but infectious disease more broadly.

## Methods

### Ethics

The study protocol and survey design were reviewed by the Institutional Review Board at Yale School of Medicine (Protocol #2000031966) and was deemed exempt from additional oversight ("Any disclosure of the human subjects' responses outside the research would not reasonably place the subjects at risk of criminal or civil liability or be damaging to the subjects' financial standing, employability, educational advancement, or reputation"). Demographic data and survey responses were only collected after the study participant had been presented with an information sheet outlining the risks and benefits of the study. Informed consent was provided by completing the survey.

### Survey

To collect data from domestic and international audiences about their opinions on the sample types used for the detection of SARS-CoV-2 (see **Table B in S1 Appendix**), we designed and circulated a survey on the online platform Qualtrics. The questionnaire contained a total of 21 questions (**see S2 Appendix for full questionnaire**). The first eight were demographic questions to obtain background information on the participants, including age, gender, education, race/ethnicity, occupation, marital status, parental status, and current country/country of origin. Questions 9 to 11 gathered information on respondents' preferences pertaining to the type of COVID-19 tests; the brand of COVID-19 tests; and what was important to them in a COVID-19 test. Next, the study participants were presented with a figure depicting and explaining the six different sample types that they were then asked about in the remaining sections of the questionnaire. Questions 12 to 14 asked which sample types (which were then explained again in text) participants had heard of; which testing options were available in the participant's area (i.e., sample type and whether PCR or rapid antigen testing); and which testing options (sample type and PCR or rapid antigen testing) participants had used. Questions 15 to 21 surveyed participants' trust of PCR and rapid antigen testing; perceptions regarding the accuracy of the different sample types; overall preference for sample type if all sampling methods were equally trustworthy and accurate; preferred sample type if they were to seek a follow-up test after a negative result; preferred sample collection method (i.e., self-collection vs. collection by a healthcare provider); preferred sample type for their children (if applicable); and, to help better inform future test development, their general preference among urine, saliva, and blood samples. All participants received the same questionnaire. The questionnaire was written in English and translated into Spanish, Dutch, French, Persian, and Arabic. The survey included both multiple-choice questions and open-ended questions. It was distributed among social networks, academic collaborators, and external laboratory partners who were encouraged to circulate the survey within their own networks. It was also distributed via email correspondence, social media channels, video conferences, newsletters, and message boards. The target survey respondent population was any individual aged 18 years and older.

### Statistical analysis

Survey results were analyzed in Python v3.11 [5] and RStudio 4.3.3 to better understand demographic differences and offer insight into public opinions on testing for leadership. For questions 9 through 11, which were open-ended, answers were translated into English using automatic detection via Google Translate. These responses were sorted into categories based on the top five most common themes detected via keywords (see **Table C in S1 Appendix**). Any answer containing a keyword in any part of the answer was sorted into the respective

category. Answers containing no response or a response not in pre-defined categories were sorted into the 'Other' category. Survey results were analyzed with pairwise Wilcoxon signed-rank tests, applying a Bonferroni correction for multiple comparisons.

## Results

### Study participants

From March 30th to June 15th, 2022, a total of 2,094 responses were collected from 7 regions: Africa (22%), Asia (8%), Europe (22%), Latin America/The Caribbean (9%), North America (27%), the Middle East (7%), and Oceania (6%) (Table 1; Table D in S1 Appendix).

**Table 1. Demographic data for survey sample.**

| Demographic | Overall (2,094) | Africa (n = 460, 22.0%) | Asia (n = 161, 7.7%) | Europe (n = 454, 21.7%) | North America (n = 556, 26.6%) | Latin America & Caribbean (n = 189, 9.0%) | Oceania (n = 122, 5.8%) | Middle East (n = 150, 7.2%) | P-value* |
|---|---|---|---|---|---|---|---|---|---|
| **GENDER** | | | | | | | | | <0.001 |
| **Female** | 1144, 54.6% | 173, 37.6% | 67, 41.6% | 280, 61.7% | 373, 67.1% | 92, 48.7% | 77, 63.1% | 82, 54.7% | |
| **Male** | 917, 43.8% | 287, 62.4% | 94, 58.4% | 168, 37.0% | 165, 29.7% | 93, 49.2% | 42, 34.4% | 66, 44% | |
| **Non-binary** | 24, 1.1% | 0, 0% | 0, 0% | 3, 0.7% | 15, 2.7% | 1, 0.5% | 3, 2.5% | 2, 1.3% | |
| **Prefer not to respond** | 9, 0.4% | 0, 0% | 0, 0% | 3, 0.7% | 3, 0.5% | 3, 1.6% | 0, 0% | 0, 0% | |
| **AGE** | | | | | | | | | <0.001 |
| **18–24** | 261, 12.5% | 111, 24.1% | 3, 1.9% | 31, 6.8% | 32, 5.8% | 7, 3.7% | 16, 13.1% | 61, 40.7% | |
| **25–34** | 566, 27.0% | 195, 42.4% | 56, 34.8% | 76, 16.7% | 131, 23.6% | 45, 23.8% | 37, 30.3% | 25, 16.7% | |
| **35–44** | 481, 23.0% | 98, 21.3% | 60, 37.3% | 84, 18.5% | 139, 25.0% | 53, 28.0% | 16, 13.1% | 31, 20.7% | |
| **45–54** | 328, 15.7% | 34, 7.4% | 27, 16.8% | 85, 18.7% | 106, 19.1% | 30, 15.9% | 25, 20.5% | 21, 14.0% | |
| **55–64** | 295, 14.1% | 18, 3.9% | 12, 7.5% | 117, 25.8% | 89, 16.0% | 33, 17.5% | 17, 13.9% | 8, 5.3% | |
| **65+** | 163, 7.8% | 4, 0.9% | 3, 1.9% | 61, 13.4% | 59, 10.6% | 21, 11.1% | 11, 9.0% | 4, 2.7% | |
| **EDUCATION LEVEL** | | | | | | | | | <0.001 |
| **Graduate degree** | 1076, 51.4% | 159, 34.6% | 137, 85.1% | 203, 44.7% | 335, 60.3% | 145, 76.7% | 38, 31.1% | 57, 38.0% | |
| **Bachelor's degree** | 553, 26.4% | 150, 32.6% | 17, 10.6% | 98, 21.6% | 164, 29.5% | 30, 15.9% | 55, 45.1% | 39, 26.0% | |
| **Some college, no degree** | 213, 10.2% | 68, 14.8% | 0, 0.0% | 68, 15.0% | 26, 4.7% | 7, 3.7% | 12, 9.8% | 32, 21.3% | |
| **OCCUPATION** | | | | | | | | | <0.001 |
| **Science / Research** | 461, 28.5% | 68, 17.8% | 34, 26.2% | 67, 21.4% | 129, 29.8% | 63, 38.7% | 60, 58.3% | 14, 15.6% | |
| **Healthcare Worker** | 436, 27.0% | 117, 30.6% | 65, 50.00% | 78, 24.9% | 65, 15.0% | 68, 41.7% | 10, 9.71% | 57, 63.3% | |
| **Other** | 719, 44.5% | 197, 51.6% | 31, 23.8% | 168, 53.7% | 239, 55.2% | 32, 19.6% | 33, 32.0% | 19, 21.1% | |

*P-values calculated using Chi-squared test.

**Table 2. Survey analytics.**

| Viewed* | Started | Completed | Drop Outs | Completion Rate | Analyzed for preference | Total Time to Complete (Average) | Total Time to Complete (Range) | Total Time to Complete (Median) |
|---|---|---|---|---|---|---|---|---|
| 5,043 | 3,694 | 2,094 | 1,600 | 67.7% | 2,094 | 15 minutes | 2.4–9,600 minutes | 7.5 minutes |

*From BITLY tracking.

Participants were 55% female, 44% male, and 1% non-binary and ranged in age from 18–24 (13%), 25–34 (27%), 35–44 (23%), 45–54 (16%), 55–64 (14%), and 65+ (7%) years. The survey had a completion rate of 68% (**Table 2**). The number of responses received for each question are detailed in **Table E in S1 Appendix**. Respondent education level was skewed, with 51% holding a graduate degree, 26% holding a bachelor's degree, and 10% with some college education. Additionally, 27% were scientists/researchers and 29% were healthcare workers.

## Awareness of sample types

The top three testing methods that participants were aware of were "using a swab to collect a sample from about halfway up your nose (anterior-nares [AN] nasal swab)" (82%), followed by "using a swab to collect a sample from going all the way through your nose to the back of your throat (nasopharyngeal swab)" (72%), then "using a swab to collect a sample from just inside your mouth (oral swab)" (62%).

## Sample type preference

Oral samples were significantly more preferred among survey respondents (**Table T in S1 Appendix**). The most preferred testing method globally was the oral swab, followed by a joint preference for the AN swab and saliva samples. The least preferred method was "deep coughing to collect fluid from in the lungs", followed by the nasopharyngeal swab (**Table 3**). However, when we asked study participants to assume that they would receive an equally accurate result from all of the sample types, "drooling saliva into a small plastic tube (saliva testing)" was the most preferred sample type globally (**Fig 1**).

## Sample type preference by region

By region (**Table 4**), the most preferred type in Africa, Europe, North America, and Oceania was the oral swab (**Table T in S1 Appendix**). In Asia, Latin America, and the Caribbean, the

**Table 3. Most preferred sample type for diagnostic testing, ranked globally.**

| Sample type | Mean Values Of Preference Rank* | P-value[#] |
|---|---|---|
| Oral swab | 2.68 | – |
| Anterior-nares (AN) nasal swab | 2.81 | 0.027 |
| Saliva | 2.86 | <0.001 |
| Oropharyngeal | 3.61 | <0.001 |
| Nasopharyngeal | 4.34 | <0.001 |
| Deep coughing | 4.62 | <0.001 |

*Lower mean, higher the preference rank.

[#]P-values calculated by Wilcoxon signed-rank test with Bonferroni correction in comparison to oral swab.

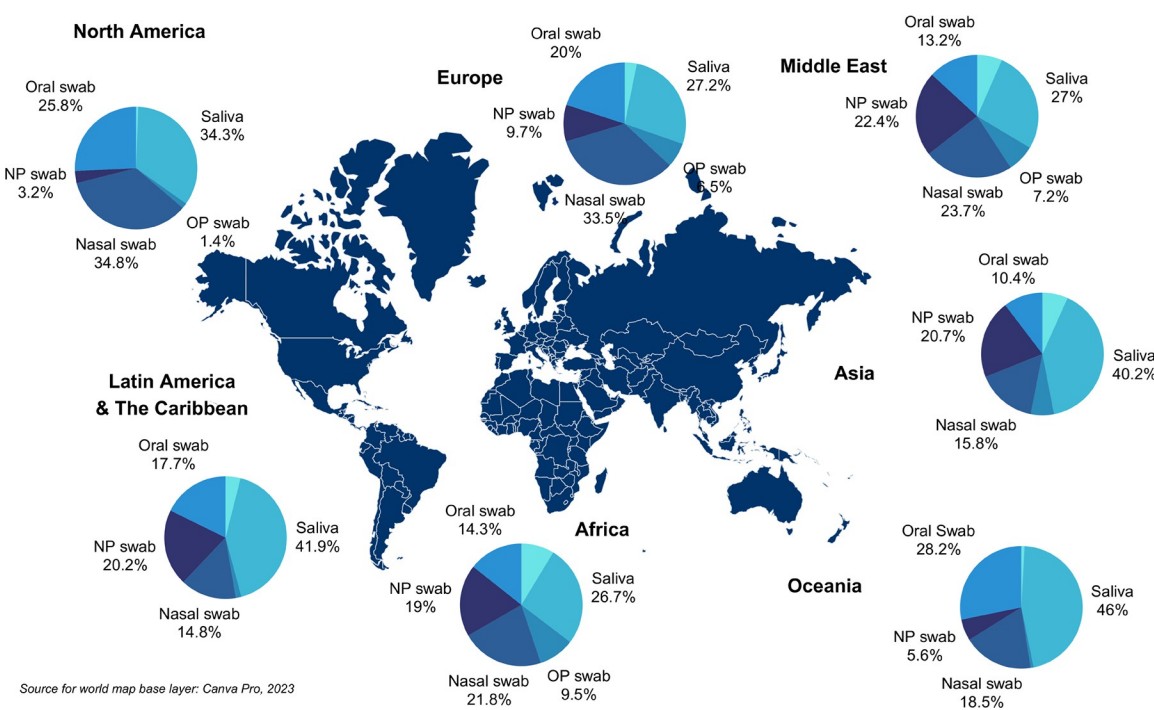

**Fig 1. Preference for diagnostic sample types by region, when study participants were asked to assume equal accuracy in testing performance.**

most preferred sample type was saliva. The most preferred sample type in the Middle East was the AN swab. Preference by country is detailed in **Tables F-L and T in S1 Appendix**.

## Sample type preference by demographic

Sample type preference varied by age group (**Table 4; Table T in S1 Appendix**); those 18–24 and over 65 preferred the AN swab, while those 25–34, 45–54, and 55–64 preferred the oral swab. Females and non-binary people preferred the oral swab, while males preferred saliva. The most preferred sample type by those holding a graduate or bachelor's degree was oral swab. In contrast, those with an associate's degree, some college education, or high school education preferred the AN swab. By race/ethnicity, the most preferred for White/Caucasian was the oral swab, for African-American was the AN swab, for Latino/Hispanic was saliva, for African and Middle Eastern was the oral swab, and for South Asian and East Asian was the AN swab. The most preferred for those in the science/research and healthcare fields was saliva, while those in administration, customer service, retail/sales, education, management, and the legal profession preferred the oral swab; those in the arts and agriculture most preferred the AN swab and oropharyngeal swab, respectively (**Table 4; Table T in S1 Appendix**).

## Qualities of test types sought after

The top three qualities most preferred by the participants in a testing method were identified as accuracy (30%), speed (21%) and ease (16%). Other sought-after qualities included sensitivity, access, and reliability. Participants also indicated that more than one of the top five qualities were important when deciding what type of test to take (20%). Additionally, 43% indicated an alternative answer (categorized as "other"), illustrating the complexity of finding a single solution or criterion for community testing programs.

**Table 4. Most preferred sample type for diagnostic testing, by demographic group.**

| Demographic | Sample Type for COVID-19 Test* | Test Administrator | Sample Type for Children | Sample Type Preference (general testing)# | Most important factors considered for COVID-19 test (in order) (>10%) |
|---|---|---|---|---|---|
| **Age Range, years (n)** | | ** | ** | | |
| **18–24 (261)** | AN Swab | Medical Professional | Saliva | Saliva | Accuracy, Speed |
| **25–34 (566)** | Oral Swab | Medical Professional | Saliva | Saliva | Accuracy, Speed, Ease, Sensitivity |
| **35–44 (481)** | Oral Swab | Medical Professional | Saliva | Saliva | Accuracy, Speed, Ease, Sensitivity |
| **45–54 (328)** | Oral Swab | Self-Collection | Saliva | Saliva | Accuracy, Speed, Ease |
| **55–64 (295)** | Oral Swab | Self-Collection | Saliva | Saliva | Accuracy, Speed, Ease |
| **65+ (163)** | AN Swab | No Preference | Oral Swab | Saliva | Accuracy, Speed, Ease |
| **Region (n, respondents)** | | ** | ** | | |
| **Asia (161)** | Oral Swab | Medical Professional | Saliva | Saliva | Accuracy, Sensitivity, Speed |
| **Europe (454)** | AN Swab | Self-Collection | Oral Swab | Saliva | Accuracy, Speed, Ease |
| **Oceania (122)** | Oral Swab | Self-Collection | Saliva | Saliva | Accuracy, Speed, Ease, Sensitivity |
| **Latin America and the Caribbean (189)** | Saliva | Medical Professional | Saliva | Saliva | Sensitivity, Speed, Accuracy |
| **North America (556)** | Oral Swab | Self-Collection | Saliva | Saliva | Accuracy, Speed, Ease |
| **Africa (460)** | Oral Swab | Medical Professional | Saliva | Saliva | Accuracy |
| **Middle East (150)** | AN Swab | Medical Professional | Saliva | Saliva | Sensitivity |
| **Gender (n, respondents)** | | ** | ** | | |
| **Male (1144)** | Saliva | Medical Professional | Saliva | Saliva | Accuracy, Speed, Sensitivity |
| **Female (917)** | Oral Swab | Medical Professional | Oral Swab | Saliva | Accuracy, Speed, Ease |
| **Non-binary (24)** | Oral Swab | Medical Professional | Oral Swab | Saliva | Accuracy |
| **Prefer not to respond (9)** | AN Swab | Medical Professional | Oral Swab | Saliva | Accuracy, Ease, Speed/Reliable |
| **Education Level (n, respondents)** | | ** | ** | | |
| **Graduate Degree (1076)** | Oral Swab | Medical Professional | Saliva | Saliva | Accuracy, Speed, Ease, Sensitivity |
| **Bachelor's Degree (553)** | Oral Swab | Self-Collection | Saliva | Saliva | Accuracy, Speed. Ease |
| **Some-college, no degree (213)** | AN Swab | Medical Professional | Oral Swab | Saliva | Accuracy, Speed, Ease |
| **Occupation (n, respondents)** | | ** | ** | | |
| **Healthcare worker (461)** | Oral Swab | Medical Professional | Saliva | Saliva | Accuracy, Sensitivity, Speed |
| **Science/Research (436)** | AN Swab | Self-Collection | Oral Swab | Saliva | Accuracy, Speed, Ease, Sensitivity |
| **Other (719)** | Oral Swab | Medical Professional | Oral Swab | Saliva | Accuracy, Speed, Ease |
| **Race (n, respondents)** | | ** | ** | | |
| **African (465)** | Oral Swab | Medical Professional | Saliva | Saliva | Accuracy |
| **South Asian (121)** | AN Swab | Medical Professional | Saliva | Saliva | Accuracy, Sensitivity, Speed |

*(Continued)*

**Table 4.** (Continued)

| Demographic | Sample Type for COVID-19 Test* | Test Administrator | Sample Type for Children | Sample Type Preference (general testing)# | Most important factors considered for COVID-19 test (in order) (>10%) |
|---|---|---|---|---|---|
| East Asian (89) | AN Swab | Medical Professional | Saliva | Saliva | Accuracy, Ease, Speed, Sensitivity |
| Latino/Hispanic (148) | Saliva | Medical Professional | Saliva | Saliva | Speed, Sensitivity, Accuracy |
| Middle Eastern (98) | Oral Swab | Medical Professional | Saliva | Saliva | Speed/Sensitivity |
| White/Caucasian (935) | Oral Swab | Self-Collection | Saliva/ AN Swab | Saliva | Accuracy, Speed, Ease |
| Other (87) | Saliva | Medical Professional | Oral Swab | Saliva | Accuracy, Speed, Sensitivity |
| Prefer not to say (35) | AN Swab | Medical Professional/ No Preference | Saliva | Saliva | Accuracy, Ease, Speed |

*Refers to preference for a COVID-19 test asked in Question 9.

#Refers to preference for more broad medical tests as asked in Question 21.

**Indicates p<0.001 by subgroup calculated by Chi-Squared test.

## Preferred sample type for children

When respondents were asked about their preferred sample type for testing their children (if applicable), respondents in Africa, Asia, North America, Latin America and the Caribbean, and Oceania favored saliva. In contrast, those in Europe favored the AN swab (**Table 4**). Preference by country is detailed in **Supplementary Tables M-S in S1 Appendix**.

## Preferred sample type for a follow-up test

The survey also asked respondents their preferred sample type for a follow-up test after receiving a negative test result but developing additional COVID-like symptoms. Across all age groups, AN swabs were the highest-ranked sample type for a follow-up test. Respondents in Africa, Europe, and North America most preferred AN swabs for a follow-up test, while those in Asia, Latin America and the Caribbean, and the Middle East preferred nasopharyngeal swabs. In Oceania, the most preferred follow-up sample type was saliva.

## Preference of other sample types

When respondents were asked to consider diagnostics testing more broadly and their preference for common liquid biopsy sample types (i.e., saliva, urine, and blood), saliva was the most preferred (72%), followed by blood (18%), then urine (9%) (**Table 5; Table U in S1 Appendix**).

**Table 5. Preference for clinical sample type ranking.**

| Clinical Sample | Overall | P-value# | Africa | Asia | Europe | North America | Latin America & Caribbean | Oceania | Middle East |
|---|---|---|---|---|---|---|---|---|---|
| Saliva | 1.37 | – | 1.64 | 1.38 | 1.36 | 1.18 | 1.31 | 1.21 | 1.47 |
| Urine | 2.22 | <0.001 | 2.20 | 2.24 | 2.16 | 2.19 | 2.33 | 2.24 | 2.46 |
| Blood | 2.40 | <0.001 | 2.15 | 2.33 | 2.44 | 2.63 | 2.38 | 2.53 | 2.08 |

#P-values calculated by Wilcoxon signed-rank test with Bonferroni correction in comparison to saliva sample.

## Discussion

Individuals previously infected with or vaccinated against SARS-CoV-2 will continue to be at risk of reinfection [6] as immunity wanes and/or as new variants continue to emerge. Moreover, global health organizations are focused on proactively planning for future pandemics and warn of the continued risk of outbreaks and emerging pathogens for which outbreak responses will be needed [7, 8]. Early detection can help mitigate the spread of infectious diseases throughout communities and improve surveillance. In order to help global experts, policymakers, and test developers create future public health measures that have public buy-in, this study sought to better understand the global community's preferences in regards to diagnostic testing, screening and surveillance, across regions, age groups, occupations, and more. Importantly, knowledge gathered on testing preferences from experiences during the COVID-19 pandemic can help better inform broader public health responses in the future.

Responses from 2,094 individuals from 125 countries in 7 regions (Africa, Europe, Asia, North America, Latin America and the Caribbean, Oceania, and the Middle East) identified a preference for oral sample types (meaning either oral swabs or saliva samples) for the detection of SARS-CoV-2, with sampling saliva as the most preferred testing method for children in 6 out of the 7 regions. There are several practical reasons which likely explain these results. First, oral sample types, including saliva, are often self-collected, and offer individuals the benefits of autonomy, convenience, and confidentiality [9]. Self-collection of any sample type may also increase testing uptake, as they are more comfortable for the patient, often faster to collect, and do not require the direct involvement of healthcare providers, thus reducing staffing, PPE, and supply pressures present for other sample types [10].

While AN swabs, oral swabs, and saliva samples are easy for individuals to self-collect, respondents in Asia, Latin America and the Caribbean, Africa, and the Middle East indicated a preference for healthcare worker-assisted sample collection. Geographic variability in self-collection preferences may stem from experiences during the COVID-19 pandemic, with different testing strategies and approvals made for self-collection in some regions but not others. For example, self-collection testing in North America was expansive and exposed individuals to various sample types, allowing them to pick their preferences. This can be seen especially in the United States with programs like the free At-Home COVID-19 Tests delivered through USPS [11], universities running large-scale self-testing programs [12], and over-the-counter testing options [13]. In contrast, countries including China predominantly relied on healthcare worker-assisted sample collection [14]. This variability in sample collection preferences should be considered by regional test developers, as well as public health officials, before designing tests and dedicating resources (monetary, personnel, etc.) to testing programs in order to maximize public acceptability.

Public health education and messaging are also critical for robust response efforts [4]. In the current study, participants responded they would prefer saliva only when we prompted them to assume that they would receive an equally accurate result from all of the sample types. Without this caveat, the most preferred testing method globally was the oral swab, followed by the AN swab. This result shows that there is a preference for saliva, yet there is doubt surrounding its accuracy. Such doubts are likely due to the mixed messaging throughout the pandemic about its reliability, which may stem from inadequate study design or interpretation [15]. Tobik *et al.* highlights that many saliva testing programs across the globe were successful [12], following its extensive validation as a reliable sample type for the detection of SARS-CoV-2 [15]. Studies have shown that when robust collection and processing methods are applied, saliva has a high concordance with nasopharyngeal swabs [12, 15], saliva-based tests are lower costs than swab-based approaches [16, 17], and that saliva is both feasible and

acceptable for routine testing at early care and education sites across the US [18, 19]. Additionally, studies from the University of Illinois [20], University of Maryland [21], and CalTech [22] all found that viral loads in saliva samples often peaked days earlier than those in nasal swabs and had a higher sensitivity during early symptom onset, indicating that saliva serves as a more reliable sample type for the early detection of SARS-CoV-2. As such, future public health and commercial efforts should focus on properly informing healthcare workers and patients on the accuracy and reliability of testing saliva so that they can have more confidence in the test being offered.

Despite potential concerns from study participants regarding the reliability of saliva for the detection of SARS-CoV-2, survey results indicated an overwhelming preference for saliva over blood and urine when thinking about broader clinical diagnostic testing. This finding suggests the potential for adopting saliva as an alternative sample type or in regions where blood sampling is common for emerging infections [23]. During outbreak response efforts, or when thinking about the development of future clinical diagnostics in general, the choice of sample types for laboratory analysis is critical [13]. Some sample types, such as blood, require skilled personnel for collection, are more costly, and may have less uptake due to peoples' aversion to needles [24]. In contrast, self-collectible samples like saliva and urine are cheaper and less invasive options requiring a fraction of the resources [24]. These lower-burden approaches are particularly well-suited to lower-resource settings. Combined with perceptions regarding the upper respiratory sample types, the acceptability of saliva may prove beneficial for diagnosing various infectious diseases, including SARS-CoV-2, MERS-CoV, Ebola, Zika [23], influenza [25], respiratory syncytial virus (RSV) [26], and more [27] in the developing world [24].

Several limitations exist in considering the broader applicability of the results in this study. With minimal resources available to support the study, we were limited to the native language abilities of those on our research team as well as close colleagues who volunteered to translate the survey for us. As such, participation in the survey would have been limited to those who could read and write in the subset we were able to provide. In addition, the survey was distributed primarily via direct outreach and word of mouth, leveraging our professional, personal, and social network contacts. Therefore, depending on their networks, responses may not represent the views and experiences of certain hard-to-reach populations, specifically those unable to access the electronic, internet-based survey. As discussed in the results section, the survey was highly skewed towards healthcare workers/researchers, who represented a combined 59.7% of the sample, and those with a graduate degree, who represented 51.4% of the survey sample. This is likely due to the reliance on direct outreach and dissemination of the survey via personal and professional networks, as the members of this research team represent this industry and level of higher education. While this may limit the generalizability of the results to the broader population, we believe these insights can still be used as a basis when considering their broader applications. Finally, our findings at the regional level do not take into account sub-regional variability. Neighboring countries that were grouped into the same region may have had vastly different access to or preferences for certain sample types, which would impact individuals' responses within that country. When designing testing programs on a global scale, researchers and public health experts should consider regional preferences and cultural factors and attitudes that may impact communities' preferences for certain testing methods.

This international survey was developed to survey the public's preferences in relation to the sample types that have been commonly used for detecting SARS-CoV-2 during the COVID-19 pandemic in an effort to better inform future public health efforts and increase test utilization. Results from this study should be considered when new testing practices are designed to encourage maximum participation from individuals which will, in turn, support overall community health. While oral sample types (either oral swabs or saliva) were preferred overall, this

survey highlighted the nuanced preferences of diagnostic testing methods across various demographic groups, especially by geographic region. By understanding where there are differences in clinical testing preferences, availability, and acceptance, future public health response efforts can be better informed and responsive to identified regional preferences.

## Supporting information

**S1 Appendix.** Table A. Types of tests for the detection of SARS-CoV-21. Table B. Sample types commonly used for the detection of SARS-CoV-2 [supporting images available upon request from the authors]. Table C. Q9—Q11 Keywords and Categories for Analysis. Table D. Geographical data for survey sample. Table E. Number of responses per question. Table F. Most preferred diagnostic testing method in Asia. Table G. Most preferred diagnostic testing method in Europe. Table H. Most preferred diagnostic testing method in Latin America & The Caribbean. Table I. Most preferred diagnostic testing method in North America. Table J. Most preferred diagnostic testing method in Africa. Table K. Most preferred diagnostic testing method in the Middle East. Table L. Most preferred diagnostic testing method in Oceania. Table M. Most preferred diagnostic testing method for children in Asia. Table N. Most preferred diagnostic testing method for children in Europe. Table O. Most preferred diagnostic testing method for children in Latin America & The Caribbean. Table P. Most preferred diagnostic testing method for children in North America. Table Q. Most preferred diagnostic testing method for children in Africa. Table R. Most preferred diagnostic testing method for children in the Middle East. Table S. Most preferred diagnostic testing method for children in Oceania. Table T. Pairwise comparison of preferred sample type for a COVID-19 test by subgroup using Wilcoxon signed-rank test with Bonferroni correction. Table U. Pairwise comparison of preferred overall clinical sample type by subgroup using Wilcoxon signed-rank test with Bonferroni correction.
(PDF)

**S2 Appendix.**
(PDF)

## Acknowledgments

We thank the study participants for their time and dedication to our study. We thank all members of the SalivaDirect Initiative and Wyllie Lab at the Yale School of Public Health for assisting in distributing the survey. In particular, we would like to thank Maikel Hislop and Yasmine Ali for their assistance with study translation. We would like to thank Craig Duni for data coding and Subhashsree Sunder for guidance on data processing. ALW would also like to thank Mike B for the many thoughtful discussions regarding this topic and survey design.

## Author Contributions

**Conceptualization:** Anne L. Wyllie.

**Data curation:** Leah Salzano, Sumaira Akbarzada, Sarah Megiel.

**Formal analysis:** Leah Salzano, Nithya Narayanan, Emily R. Tobik, Yanjun Wu.

**Funding acquisition:** Anne L. Wyllie.

**Investigation:** Anne L. Wyllie.

**Methodology:** Leah Salzano, Brittany Choate, Anne L. Wyllie.

**Project administration:** Anne L. Wyllie.

**Supervision:** Brittany Choate, Anne L. Wyllie.

**Visualization:** Nithya Narayanan, Brittany Choate.

**Writing – original draft:** Leah Salzano, Nithya Narayanan, Emily R. Tobik, Sumaira Akbarzada, Brittany Choate, Anne L. Wyllie.

**Writing – review & editing:** Leah Salzano, Nithya Narayanan, Emily R. Tobik, Sumaira Akbarzada, Yanjun Wu, Sarah Megiel, Brittany Choate, Anne L. Wyllie.

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
