## [Decision Letter · Decision Letter 0]

9 Apr 2024

PGPH-D-23-02560

Diagnostic testing preferences can help inform future public health response efforts: global insights from an international survey

Dear Dr. Wyllie,

Thank you for submitting your manuscript to PLOS Global Public Health. After careful consideration, we feel that it has merit but does not fully meet PLOS Global Public Health’s publication criteria as it currently stands. Therefore, we invite you to submit a revised version of the manuscript that addresses the points raised during the review process. Please address each points raised by the reviewers separately in your reply with the two critical aspects outlined below.

We look forward to receiving your revised manuscript.

Kind regards,

Nei-yuan (Marvin) Hsiao

Academic Editor

Journal Requirements:

2. Please send a completed 'Competing Interests' statement, including any COIs declared by your co-authors. If you have no competing interests to declare, please state "The authors have declared that no competing interests exist". Otherwise please declare all competing interests beginning with the statement "I have read the journal's policy and the authors of this manuscript have the following competing interests:"

3. Please amend your detailed Financial Disclosure statement. This is published with the article. It must therefore be completed in full sentences and contain the exact wording you wish to be published.

If you did not receive any funding for this study, please simply state: “The authors received no specific funding for this work.

4. Please provide separate figure files in .tif or .eps format only and remove any figures embedded in your manuscript file. Please also ensure all files are under our size limit of 10MB.

5. We notice that your supplementary files are included in the manuscript file. Please remove them and upload them with the file type 'Supporting Information'. Please ensure that each Supporting Information file has a legend listed in the manuscript after the references list.

6. Some material included in your submission may be copyrighted. According to PLOS’s copyright policy, authors who use figures or other material (e.g., graphics, clipart, maps) from another author or copyright holder must demonstrate or obtain permission to publish this material under the Creative Commons Attribution 4.0 International (CC BY 4.0) License used by PLOS journals. Please closely review the details of PLOS’s copyright requirements here: PLOS Licenses and Copyright. If you need to request permissions from a copyright holder, you may use PLOS's Copyright Content Permission form.

Potential Copyright Issues:

Fig 1: please (a) provide a direct link to the base layer of the map (i.e., the country or region border shape) and ensure this is also included in the figure legend; and (b) provide a link to the terms of use / license information for the base layer image or shapefile. We cannot publish proprietary or copyrighted maps (e.g. Google Maps, Mapquest) and the terms of use for your map base layer must be compatible with our CC-BY 4.0 license. 

Additional Editor Comments (if provided):

Please ensure the following points raised by reviewers are adequately addressed in the revision/rebuttal:

1. Clarification and discussion of sampling strategy which may impact the finding of the survey (reviewer 2).

2. Around the use of the term "saliva", "oral sample" and "oral swab", they do not appear to be properly defined in the survey questions and there are questions from reviewers whether responders understood the implication of these term. Please define them clearly in the manuscript and outline the researchers' attempt to clarify this in the survey. Finally, it is important to discuss the limitations of this accordingly as it is one of the primary purpose of the study.

Reviewers' comments:

Reviewer's Responses to Questions

**Comments to the Author**

1. Does this manuscript meet PLOS Global Public Health’s publication criteria? Is the manuscript technically sound, and do the data support the conclusions? The manuscript must describe methodologically and ethically rigorous research with conclusions that are appropriately drawn based on the data presented.

Reviewer #1: Yes

Reviewer #2: Yes

2. Has the statistical analysis been performed appropriately and rigorously?

Reviewer #1: No

Reviewer #2: Yes

3. Have the authors made all data underlying the findings in their manuscript fully available (please refer to the Data Availability Statement at the start of the manuscript PDF file)?

Reviewer #1: Yes

Reviewer #2: Yes

4. Is the manuscript presented in an intelligible fashion and written in standard English?

Reviewer #1: Yes

Reviewer #2: Yes

5. Review Comments to the Author

Reviewer #1: Salzano et al. conducted research to determine the preferred type of sample during the COVID-19 pandemic among individuals living in various regions across the globe. This study holds great importance and provides valuable insights for better preparedness in the face of future pandemics. Below are my observations.

• Terms such as testing preference and testing samples are not clear. What does testing preference mean? Is it a type of specimen or a type of lab method? The manuscript needs revision in this regard.

• Abstract: Oral sample versus oral swab, are they the same? I think it should be an oral swab instead of an oral sample.

• Line number 89 to 91: “ We aimed to 90 better understand the public view regarding different sample types with the hope that these results could 91 inform future surveillance planning and diagnostics development. Surveillance for which disease? Is it for COVID? Further description is needed.

• It is advisable to expand all abbreviations in the first encounter.

• Would you clarify the ‘testing method’ mentioned in Tables 3 and 4? I did not see any testing method in the table.

• Add some statistical analysis

Reviewer #2: The manuscript presents a significant and timely contribution to the field of public health and diagnostics, focusing on global preferences for clinical sample types used in SARS-CoV-2 detection. The study's relevance is underscored by the ongoing challenges in managing infectious diseases and the imperative for testing strategies that are both effective and acceptable to the public. Below are comments aimed at further strengthening the manuscript:

1. While the broad geographic distribution of the survey is a strength, more detail on the survey distribution methods would enhance the reader's understanding of potential sampling biases. Specifically, clarifying how reliance on social networks and professional contacts might have skewed demographics towards healthcare professionals and those with higher education levels would be informative.

2. The discussion on the reliability and public perception of saliva testing could be strengthened by incorporating more comparative data or studies on its efficacy relative to other methods.

3. The manuscript could benefit from a deeper examination of the implications of self-collected samples on testing logistics, healthcare system burden, and potential barriers to access.

4. Clarify how the criteria for choosing the translated languages and how this selection might have influenced regional participation rates.

Addressing the points above could enhance the clarity, rigor, and applicability of the work.

6. PLOS authors have the option to publish the peer review history of their article (what does this mean?). If published, this will include your full peer review and any attached files.

**Do you want your identity to be public for this peer review?** For information about this choice, including consent withdrawal, please see our Privacy Policy.

Reviewer #1: No

Reviewer #2: No

---

## [Editor Report · Decision Letter 1]

10 Jul 2024

Diagnostic testing preferences can help inform future public health response efforts: global insights from an international survey

PGPH-D-23-02560R1

Dear Dr Wyllie,

We are pleased to inform you that your manuscript 'Diagnostic testing preferences can help inform future public health response efforts: global insights from an international survey' has been provisionally accepted for publication in PLOS Global Public Health.

Best regards,

Nei-yuan (Marvin) Hsiao

Academic Editor